

# Monte Carlo simulation of OLS and linear mixed model inference of phenotypic effects on gene expression

Jeffrey A. Walker

Department of Biological Sciences, University of Southern Maine, Portland, ME, United States

## ABSTRACT

**Background**. Self-contained tests estimate and test the association between a phenotype and mean expression level in a gene set defined *a priori*. Many self-contained gene set analysis methods have been developed but the performance of these methods for phenotypes that are continuous rather than discrete and with multiple nuisance covariates has not been well studied. Here, I use Monte Carlo simulation to evaluate the performance of both novel and previously published (and readily available via R) methods for inferring effects of a continuous predictor on mean expression in the presence of nuisance covariates. The motivating data are a high-profile dataset which was used to show opposing effects of hedonic and eudaimonic well-being (or happiness) on the mean expression level of a set of genes that has been correlated with social adversity (the CTRA gene set). The original analysis of these data used a linear model (GLS) of fixed effects with correlated error to infer effects of *Hedonia* and *Eudaimonia* on mean CTRA expression.

**Methods**. The standardized effects of *Hedonia* and *Eudaimonia* on CTRA gene set expression estimated by GLS were compared to estimates using multivariate (OLS) linear models and generalized estimating equation (GEE) models. The OLS estimates were tested using O'Brien's OLS test, Anderson's permutation $r_F^2$-test, two permutation $F$-tests (including GlobalAncova), and a rotation $z$-test (Roast). The GEE estimates were tested using a Wald test with robust standard errors. The performance (Type I, II, S, and M errors) of all tests was investigated using a Monte Carlo simulation of data explicitly modeled on the re-analyzed dataset.

**Results**. GLS estimates are inconsistent between data sets, and, in each dataset, at least one coefficient is large and highly statistically significant. By contrast, effects estimated by OLS or GEE are very small, especially relative to the standard errors. Bootstrap and permutation GLS distributions suggest that the GLS results in downward biased standard errors and inflated coefficients. The Monte Carlo simulation of error rates shows highly inflated Type I error from the GLS test and slightly inflated Type I error from the GEE test. By contrast, Type I error for all OLS tests are at the nominal level. The permutation $F$-tests have ∼1.9X the power of the other OLS tests. This increased power comes at a cost of high sign error (∼10%) if tested on small effects.

**Discussion**. The apparently replicated pattern of well-being effects on gene expression is most parsimoniously explained as "correlated noise" due to the geometry of multiple regression. The GLS for fixed effects with correlated error, or any linear mixed model for estimating fixed effects in designs with many repeated measures or outcomes, should be used cautiously because of the inflated Type I and M error. By contrast, all OLS

Corresponding author
Jeffrey A. Walker, walker@maine.edu

tests perform well, and the permutation $F$-tests have superior performance, including moderate power for very small effects.

## INTRODUCTION

The motivation for this work is a recent self-contained test, a type of gene set analysis (*Goeman & Bühlmann, 2007*), that used a linear model (GLS) with correlated error to estimate the mean effect of a phenotype on a set of expression levels for a gene set identified *a priori* (*Fredrickson et al., 2015*). The development of self-contained tests is an active area of research in genomics (*Tian et al., 2005*; *Goeman & Bühlmann, 2007*; *Hummel, Meister & Mansmann, 2008*; *Wu et al., 2010*; *Tripathi & Emmert-Streib, 2012*; *Zhou, Barry & Wright, 2013*). More generally, the effect of a phenotype on the mean response of multiple outcomes, has a long and rich history in applied statistics, especially in the context of clinical outcomes in medicine (*O'Brien, 1984*; *Pocock, Geller & Tsiatis, 1987*; *Lauter, 1996*; *Bull, 1998*). The development of self-contained test methods within the field of genomics has advanced largely without reference to this earlier literature (but see *Chen et al., 2007*; *Tsai & Chen, 2009*). The use of GLS in self-contained tests of genomics data has not been widely used (if at all) outside of *Fredrickson et al. (2015)* and subsequent papers by the same authors. Results presented by *Fredrickson et al. (2015)* raise several concerns about the stability of the estimates and their standard errors, especially given the well known problems with the GLS model when used for repeated measures or multiple outcome data (*Littell et al., 2006*).

Here, I re-analyze the data of *Fredrickson et al. (2015)*, and an earlier "discovery" dataset (*Fredrickson et al., 2013*), with two goals in mind. First, and more specifically, I re-analyze the datasets using alternatives to GLS for testing for fixed effects on multiple responses. The alternatives include O'Brien's OLS test for multiple outcomes (*O'Brien, 1984*), a permutation $r_F^2$-test (*Anderson & Robinson, 2001*), two different permutation partial $F$-tests, including GlobalAncova (*Hummel, Meister & Mansmann, 2008*), a rotation $z$-test (Roast) (*Wu et al., 2010*), and a Wald test using robust standard errors (*Zeger & Liang, 1986*). Second, and more generally, I use Monte Carlo simulation experiments to evaluate the performance of the GLS and alternative methods when used on simulated data explicitly modeled on the dataset from *Fredrickson et al. (2015)*. By simulating a specific dataset, I can compare test methods without extrapolating from more general, simulation studies. Importantly, the empirical and simulated data contain numerous nuisance covariates, and the focal predictors are continuous rather than discrete, and, consequently, these data are more complex than the data found in most previous comparisons of gene-set association methods.

I show that all OLS tests have a Type I error very close to the nominal rate, but that the two permutation tests that use an $F$-ratio as the test-statistic have superior power.

However, O'Brien's OLS test, which doesn't require permutation, is extremely fast and may be useful for competitive gene-set association tests that require many more tests.

## Background

*Fredrickson et al. (2015)* reported a large, negative, highly statistically significant, standardized effect (partial regression coefficient) of eudaimonic well-being on the "conserved transcriptional response to adversity" (CTRA) gene set (*Fredrickson et al., 2015*). Additionally, using OLS estimates, the authors reported opposing effects of eudaimonic and hedonic well being on CTRA expression (*Fredrickson et al., 2015*), a pattern that they argue replicates the results of an earlier 2013 study (*Fredrickson et al., 2013*). Hedonia and eudaimonia are two types of well-being that are variously defined in the psychological literature but, effectively, hedonia is the striving for or feeling of pleasure and absence of distress while eudaimonia is the striving for or feeling of betterment and meaningfulness (*Huta & Waterman, 2014*). I use italics (*Hedonia* and *Eudaimonia*) to indicate the empirical measures of hedonia and eudaimonia in the analyzed datasets. The 2015 study (*Fredrickson et al., 2015*) was a replication of an earlier study that reported not only a negative effect of *Eudaimonia* on CTRA expression but also a positive effect of *Hedonia* on CTRA expression (*Fredrickson et al., 2013*). While the 2015 study did not report a statistically significant effect of hedonic well-being on CTRA expression, it did report plots showing opposing effects of hedonic and eudaimonic scores on mean CTRA gene expression that were in the same direction as that in *Fredrickson et al. (2013)*. This apparent replication of opposing effects was again emphasized in a published correction (*Fredrickson et al., 2016*).

The CTRA gene set includes 19 pro-inflammatory, 31 anti-viral, and 3 antibody-stimulating genes. The *Fredrickson et al. (2013)* data included all 53 genes but the *Fredrickson et al. (2015)* data is missing IL-6 from the pro-inflammatory subset. *Fredrickson et al. (2013)* used 53 univariate multiple regressions to estimate the effects (the regression coefficient) of each well-being (hedonic and eudaimonic) score on log2 (normalized gene expression) for each gene. The regression model included both well-being scores, seven covariates to adjust for demographic and general health confounding (sex, age, ethnicity, body-mass index, and measures of alcohol consumption, smoking, and recent illness), and eight expression levels of T-lymphocyte markers to adjust for immune status confounding. Hedonic and eudaimonic scores were transformed to $z$-scores prior to the analysis. The 53 multiple regressions (one for each gene) yielded 53 coefficients for hedonic score and 53 coefficients for eudaimonic score. The coefficients of the 31 anti-viral and three antibody genes were multiplied by $-1$ to make the direction of the effect consistent with the CTRA response. *Fredrickson et al. (2013)* used a simple one-sample $t$-test of the 53 coefficients to test for a mean effect of hedonic or eudaimonic score on CTRA expression. A mean coefficient greater than zero reflects a positive CTRA response (increased pro-inflammatory and decreased anti-viral and antibody-stimulating genes).

*Fredrickson et al. (2013)* used a bootstrap to re-sample the coefficients in order to generate a standard error (the denominator of their $t$-value) and then tested the statistic using $m - 1$ degrees of freedom, where $m$ is the number of outcomes (gene expression levels). There are two fundamental problems with this $t$-test. First, the coefficients are not

independent of each other because of the correlated expression levels among genes and as a consequence the standard error in the denominator will be too small, which should result in an inflated Type I error rate. Second, their degrees of freedom does not account for the number of subjects in the study. At the extreme, if only a single gene expression level is measured, Fredrickson's $t$ cannot even be computed. This second error should result in loss of power. The combined effect on Type I and Type II error will depend on the magnitude of the correlations among the expression levels. Through simulation, however, *Brown et al. (2014)* discovered an inflated Type-I error in their exploration of the data using the *Fredrickson et al. (2013)* $t$-test. *O'Brien (1984)* developed an appropriate $t$-test for the effects of an an independent variable on multiple outcomes (see below).

*Fredrickson et al. (2015)* replicated the 2013 study but treated the 52 gene expression levels as "repeated" measures (or multiple outcomes) of a single expression response and used a linear model with fixed-effects and correlated error to estimate the regression coefficients of expression on hedonic and eudaimonic score. Specifically, *Fredrickson et al. (2015)* used generalized least squares (GLS) with a heterogenous compound symmetry error matrix to estimate the marginal (population-averaged) fixed effects. Compound symmetry assumes equal correlation (conditional on the set of predictors) among all expression levels. This is not likely to approximate the true error structure for a set of expression levels for different genes, as these expression levels will share different sets of underlying regulatory factors. *Fredrickson et al. (2015)* re-ran the analysis using an unstructured error matrix, with results contradicting the compound symmetry results, but chose to report this in the supplement and not the main text. Regardless, linear mixed models for repeated measures or multiple outcomes are prone to inflated Type I error due to both upward biased effect estimates and downward biased standard errors (*Kackar & Harville, 1984*; *Kenward & Roger, 1997*; *Guerin & Stroup, 2000*; *Littell et al., 2006*; *Jacqmin-Gadda et al., 2007*; *Gurka, Edwards & Muller, 2011*). The amount of bias depends on the true and specified correlation structure, as well as effective sample size (a function of the number of subjects, the number of outcomes, and the correlations among the outcomes), but can be large even with large samples (*Gurka, Edwards & Muller, 2011*). When only the marginal effects are of interest (as here), population-averaged effects are typically estimated using Generalized Estimating Equations (GEE) instead of GLS and standard errors robust to model misspecification are computed using the sandwich estimator (*Liang & Zeger, 1986*; *Zeger & Liang, 1986*).

Finally, *Cole et al. (2015)* replicated the 2015 study, using the same GLS method, but only reported the effect of *Eudaimonia* and not *Hedonia* on CTRA expression. By contrast to the earlier study, an unstructured error matrix was specified and the expression levels were not standardized to $z$-scores. Again, a negative effect of *Eudaimonia* on CTRA expression was reported but the magnitude cannot be easily compared to other values because of the lack of standardization.

## METHODS

Data from *Fredrickson et al. (2013)* (hereafter FRED13) and *Fredrickson et al. (2015)* (hereafter FRED15) were the primary datasets re-analyzed because both contain the well-being

measure *Hedonia* in addition to *Eudaimonia*. A more abbreviated analysis of the data from *Cole et al. (2015)*, which did not include a measure of *Hedonia*, was performed simply to show the inconsistency of the GLS method when used consistently among datasets. Data were downloaded as .txt Series Matrix Files from http://www.ncbi.nlm.nih.gov/geo/ using accession numbers GSE45330 (the 2013 dataset, hereafter FRED13), GSE55762 (the focal 2015 dataset, hereafter FRED15) and GSE68526 (the replicate 2015 dataset, hereafter COLE15). The CTRA (response) expression data were log2 transformed. The T-lymphocyte expression data that formed part of the set of covariates were log2 transformed in the downloaded data. The downloaded hedonic and eudaimonic scores in FRED13 had means and variances close but not equal to that expected of *z*-scores, which suggests that the public data slightly differs from that analyzed by *Fredrickson et al. (2013)*; these were re-standardized to *z*-scores Three rows of FRED13 had missing covariate data (two rows were completely missing) and were excluded; the number of rows (subjects) in the cleaned matrix was 76. The downloaded hedonic and eudaimonic scores in FRED15 were the raw values and were transformed to *z*-scores. There was no missing data in FRED15 and the number of subjects was 122. The COLE15 data did not include a measure for *Hedonia*. Excluding rows with missing values left complete data for 108 subjects.

Prior to all analyses, *Hedonia* or *Eudaimonia* scores and the expression levels of all genes were standardized to mean zero and unit variance. Additionally, the 31 anti-viral and 3 antibody genes were multiplied by $-1$ to make the direction of the effect consistent with the CTRA response (*Fredrickson et al., 2013*; *Fredrickson et al., 2015*).

## Null hypothesis tests

If $\beta_j$ is the effect (partial regression coefficient) of *Hedonia* or *Eudaimonia* on the expression level of the *j*th gene, the overall effect of *Hedonia* or *Eudaimonia* on expression of the CTRA gene set is simply the averaged coefficient over all genes, $\bar{\beta} = \frac{1}{m}\sum \beta_j$ where *m* is the number of genes. The three focal null hypotheses are $H_0 : \bar{\beta}_{hedonia} = 0$, $H_0 : \bar{\beta}_{eudaimonia} = 0$, and $H_0 : \delta_{\text{hed}-\text{eud}} = \bar{\beta}_{hedonia} - \bar{\beta}_{eudaimonia} = 0$. All three hypotheses are directional; that is, the mean effect differs from zero. This differs from the general multivariate test that at least one of the coefficients differs from zero, but the mean response may be zero. While the hypotheses are directional, the tests are two-tailed, that is, the mean response may be up or down regulation of the CTRA gene set.

## OLS inferential tests

The effects of *Hedonia* and *Eudaimonia* on the mean of the *m* gene expression levels are estimated with the multivariate linear model

$$Y = XB + E \tag{1}$$

where $Y$ is the $n \times m$ matrix of gene expression levels for the *n* subjects, $X$ is the model matrix of dummy variables and covariates, $E$ is the matrix of residual error, and $B$ is the $p \times m$ matrix of partial regression coefficients. For the combined data, the model matrix includes a dummy variable indicating dataset (2013 or 2015). The coefficients of the *j*th column of $B$ are precisely equal to those from a univariate multiple regression

of the $j$th gene on $X$ (and why the model is sometimes called a multivariate multiple regression). In R, estimating the $m$ effects of *Hedonia* and *Eudaimonia* is much faster using this multivariate model than looping through $m$ univariate multiple regressions. The mean of the $m$ coefficients is the OLS estimate of the effect of *Hedonia* or *Eudaimonia* on overall CTRA expression level. Because the well-being scores for *Hedonia* and *Eudaimonia* and the $m$ expression levels were mean-centered and variance-standardized, the reported OLS estimates are mean (averaged over the $m$ genes) standard partial regression coefficients.

### O'Brien's OLS t-test

O'Brien's OLS test (*O'Brien, 1984*; *Logan & Tamhane, 2004*; *Dallow, Leonov & Roger, 2008*) was developed explicitly for testing the directional hypothesis that the mean effect of multiple outcomes differs from zero, which is precisely the question pursued in both Fredrickson papers. Given $m$ standardized regression coefficients and associated $t$-values, O'Brien's test statistic is

$$t_{\text{Obrien}} = \frac{j^\top t}{\sqrt{j^\top R j}} \tag{2}$$

where $j$ is a $m$ vector of 1s, $t$ contains the $t$-values associated with each of the $m$ partial regression coefficients, and $R$ is the conditional correlation matrix of the $m$ expression levels. $R$ was computed separately for *Hedonia* and *Eudaimonia* from the residuals of the multivariate linear model (Eq. (1)) with all covariates in the model except the focal covariate (the reduced model). A $t$ distribution with $n - p - 1$ degrees of freedom (where $p$ is the number of predictors) was used to estimate the two-tail probability of $|t| \geq |t_{\text{Obrien}}|$ given the null. A standard error of the mean effect was computed as $SE = \frac{\bar{\beta}}{t_{\text{Obrien}}}$

### Anderson's permutation $r_F^2$-test

As an alternative to the parametric O'Brien's OLS test, I use four different permutation, or permutation-like tests. The first of these is Anderson's permutation $r_F^2$-test of the partial correlation between the focal predictors ($Z$) and the outcomes ($Y$) conditional on the covariates ($X$) (any of these can be univariate or multivariate) (*Anderson & Robinson, 2001*; *Anderson, 2001*). This test uses a permutation of the residuals of the null model ("permutation under the null") as these residuals, but not the $Y$, $X$, or $Z$, are exchangeable. *Freedman & Lane (1983)*, initially developed the permutation under the null using a $t$-value of the effect (for a gene set with $m > 1$, this statistic would be the mean or sum of the $t$-values for each gene). Anderson provided both theoretical and empirical evidence for the superior performance of the permutation under the null, but used the squared partial correlation coefficient $r_F^2 = \rho_{zy.x}^2$ in place of $t$. For a permutation under the null, the predictor variables are divided into main effects $Z$ (hedonic or eudaimonic scores were tested independently) and nuisance covariates $X$ (the demographic and immune variables plus the well-being score not being tested) and the model becomes

$$Y = XA + ZB + E \tag{3}$$

and the two-sided test statistic is

$$r_F^2 = \frac{(\sum \boldsymbol{E}_{\pi(F)} \boldsymbol{E}_{Z|X})^2}{\sum \boldsymbol{E}_{\pi(F)}^2 \sum \boldsymbol{E}_{Z|X}^2} \tag{4}$$

where $\boldsymbol{E}$ are the residuals from different fit models. The computations for this are

1. Compute $\boldsymbol{E}_{Z|X}$, which are the residuals of $\boldsymbol{Z}$ regressed on $\boldsymbol{X}$.
2. Set $\boldsymbol{B} = 0$ and fit model 3. The residuals from this model are the estimated residuals under the null ($\boldsymbol{E}_{Y|X}$) and the predicted values are the estimated $Y$ under the null ($\hat{\boldsymbol{Y}}$).
3. Permute the residuals under the null and add to the predicted values under the null, $\boldsymbol{Y}_\pi = \hat{\boldsymbol{Y}} + \boldsymbol{E}_{Y|X}^\pi$, where $\pi$ indicates permutation.
4. Fit model model 3, again with $\boldsymbol{B} = 0$ but substitute $\boldsymbol{Y}_\pi$ for $\boldsymbol{Y}$, and compute the residuals from this fit, which are the $\boldsymbol{E}_{\pi(F)}$.

   To generate the null distribution of the test statistic, 10,000 permutations were run, including an iteration of non-permuted data. The two-sided $p$-value of each hypothesis was computed as the fraction of $r_F^2 \geq$ the observed $r_F^2$.

### Permutation $F_{ga}$-test (GlobalAncova)

Because it is implemented in the GlobalAncova package (*Mansmann et al., 2010*), the permutation $F$-test described in *Hummel, Meister & Mansmann (2008)* is an attractive alternative to the permutation $r_F^2$-test. The GlobalAncova test statistic, $F_{ga}$ compares the residual sums of squares of the reduced model not including the predictor(s) of interest ($Z$) to the residual sums of squares of the full model. The full model 3 is fit but substituting the residuals of the reduced model ($\boldsymbol{E}_{Y|X}^\pi$) for $\boldsymbol{Y}$. The GlobalAncova test with 10,000 permutations was run separately for *Hedonia* and *Eudaimonia*. GlobalAncova permutes the main effects ($Z$), and thus does not preserve the covariance relationship between the main effects and the nuisance effects. While the $Z$ are exchangeable under the null in experimental designs when they are assigned randomly, they are not exchangeable under the null in observational designs (*Freedman & Lane, 1983*; *Anderson, 2001*). The Monte Carlo simulation results for data modeled on FRED15 (see below) suggest that the violation of exchangeability is trivial.

### Permutation $F_{pun}$-test

The GlobalAncova $F_{ga}$-test had high power with these data (see 'Results') but a concern was whether this high power was due to the violation of exchangeability under the null. Consequently, I implemented a modification of the GlobalAncova test by permuting the residuals under the null to generate a permuted response ($\boldsymbol{Y}_\pi$) (see above description for the permutation $r_F^2$-test). Each iteration, the full models and reduced models were fit to $\boldsymbol{Y}_\pi$ and the respective residual sums of squares were computed (note that in GlobalAncova, the reduced-model residual sums of squares are only computed once, for the observed data). I refer to this as the permutation $F_{pun}$-test (for "permutation under the null"). A total of 10,000 permutations were run, including an iteration of non-permuted data. The $p$-value of each hypothesis was computed as the fraction of $F_{pun} \geq$ the observed $F_{pun}$.

### Rotation z-test (ROAST)

The rotation-test described in *Wu et al. (2010)* is an attractive alternative to the permutation tests as it is small-sample exact and is implemented in the function roast from the limma package (*Ritchie et al., 2015*) (by contrast, permutation tests are only asymptotically exact). The test statistic, $z_{\text{rot}}$, is a mean $z$-score computed from the set of $m$ moderated $t$-statistics computed for each gene. Using a hierarchical model, the moderated $t$-statistic uses information on the error of all genes in the set to estimate the gene specific standard error. A $p$-value for the test statistic is evaluated in a very similar manner to the permutation tests described above, but, instead of permutation, the $n$-vector of reduced-model residuals is rotated by a random vector $r$, which is constant for all genes within each iteration but variable among iterations. The observed and rotated $z$-scores from 10,000 rotations were used to generate the null distribution. The $p$-value for the "UpOrDown" test was used as this is the test of the two-tailed directional hypothesis.

## Inference using linear model with fixed effects and correlated error

The model used by *Fredrickson et al. (2015)* is

$$y_i = X_i \beta + \varepsilon_i \tag{5}$$

where $y_i$ is the vector of $m$ responses for subject $i$, $X_i$ is the model matrix for subject $i$, which includes the main effect *Gene* to identify the $j$th element of $y_i$, and $\beta$ is the vector of fixed (or population-averaged) effects. In this model, $\varepsilon_i \sim N(0, \Sigma)$, where $\Sigma$ is the within subject error covariance matrix representing the correlated errors. The correlated errors result from random effects due to subjects but the model does not explicitly model these. To implement this model, the data matrix with separate columns for each gene is stacked into long format by combining the $m$ expression levels into a single response variable (*Expression*) and the variable *Gene* is created to identify the gene associated with a specific response (expression value). The univariate regression of *Expression* on the set of predictors results in the same OLS estimates as in the multivariate model described above. These estimates are unbiased but the standard errors for the estimates are incorrect because of the correlated errors. As in the multivariate model for the combined data, the model matrix includes a dummy variable indicating dataset (2013 or 2015).

### Generalized estimating equations

*Fredrickson et al. (2015)* used maximum likelihood with the GLS model with a heterogenous compound symmetry error matrix to estimate the fixed effects in Eq. (5). To replicate these results, I estimated the fixed effects and their standard errors and $p$-values using the gls function from the nmle package (*Pinheiro et al., 2015*) (with a heterogenous compound symmetry error matrix and using maximum likelihood method). Additionally, because only the fixed effects are of interest, and because of the known bias in the standard errors of the GLS with correlated errors model, I used Generalized Estimating Equations (GEE) with an exchangeable error matrix to estimate the fixed effects using the function geeglm in the geepack package (*Yan, 2002*). The default sandwich estimator was used to compute the standard errors of the effects, which is robust to error covariance misspecification (*Liang*

& Zeger, 1986). Nevertheless, GEE is less efficient if the error covariance is misspecified (Sammel & Ryan, 2002).

### Permutation and bootstrap GLS

Exploration of the behavior of the GLS as implemented by *Fredrickson et al. (2015)* suggested partial regression coefficients that were more unstable than implied by the standard error. To explore the consequences of this instability on inference, I implemented both a bootstrap procedure to compute approximate standard errors and the Freedman and Lane permutation procedure (*Anderson & Robinson, 2001*) described above to compute permutation-GLS p-values. Each iteration of either the bootstrap or the permutation, the data were resampled (or the residuals permuted) in wide format, rescaled, and reshaped to long format. Coefficients were estimated using the gls function as described above. The first iteration used the observed (not resampled) data. The standard partial regression coefficients and associated $t$-values for *Hedonia* and *Eudaimonia* were saved each iteration and used to generate standard errors for the bootstrap and a null distribution of $t$-values for the permutation. Because the time required to fit the GLS, and the exploratory goal of this analysis, I used 200 samples for the bootstrap and 400 samples for the permutation, which are sufficient for approximate values. The regressor *Smoke* was excluded from the bootstrap analysis because some bootstrap samples had zero cases with level $Smoke = 1$, which leads to an unsolvable model. I show in the results that this exclusion of *Smoke* has trivial effects on the estimates of the coefficients and standard errors.

## Monte Carlo simulations of errors

I used Monte Carlo simulation to explore Type I, Type II, Type S, and Type M errors with data similar in structure to the focal FRED15 dataset. Type S and Type M error are the errors in the sign and magnitude of the estimated coefficient when $p < 0.05$ (*Gelman & Carlin, 2014*). For Type S error, I used the (frequentist) probability that the sign of the coefficient is wrong when $p < 0.05$, which is $\frac{N(p<0.05|\beta_{estimated}<0,\beta_{true}>0)}{N(p<0.05)}$, where $N(x)$ is the number of iterations with condition $x$ (*Gelman & Carlin, 2014*). For Type M error, I used the exaggeration ratio ($\frac{\beta_{estimated}}{\beta_{true}}$) (*Gelman & Carlin, 2014*). In each run of the simulation, a random $n \times p$ matrix $\mathbf{X}$ of independent variables ($n$ samples of $p$ covariates) and a random $n \times m$ matrix $\mathbf{Y}$ of response variables ($n$ samples of $m$ responses) were generated using the function rmvnorm from the mvtnorm package (*Genz et al., 2015*). All simulated independent variables were modeled as continuous variables sampled from $\mathcal{N}(\mathbf{0}, \mathbf{S}_X)$, where $\mathbf{S}_X$ is the covariance matrix of the 17 regressor variable from FRED15. The 52 response variables were modeled as continuous variables sampled from $\mathcal{N}(\mathbf{0}, \mathbf{S}_Y)$, where $\mathbf{S}_Y$ is the covariance matrix of the 52 gene expression levels from FRED15. For the power simulations (including Type S and M errors), the standardized effect of *Eudaimonia* on the mean response was set to 0.067, which is the estimated, standardized effect for the FRED15 dataset. The effect of all other covariates, including that of *Hedonia* was set to zero. For the Type I simulations, all effects were set to zero. Sample size (the number of subjects $n$) was set to that for FRED15 (122) for all runs. To explore the consequences of increasing the gene set size on error rates, the simulation was run with three levels of $m$ (10, 30, 52). The

**Table 1  GLS estimates of the variance-standardized coefficients for the 2013, 2015, and combined data.** The GLS bootstrap SE and permutation $p$-values are also given. $\delta_{hed-eud}$ is the difference in the estimates: $\beta_{hedonia} - \beta_{eudaimonia}$.

| Type | Data | Estimate | SE | $p$ | $SE_{boot}$ | $p_{gls-perm}$ |
|------|------|----------|-----|-----|-------------|----------------|
| *Hedonia* | FRED13 | 0.537 | 0.172 | 0.002 | 0.664 | 0.29 |
| | FRED15 | 0.086 | 0.122 | 0.484 | 0.296 | 0.74 |
| | FRED13 + 15 | 0.073 | 0.042 | 0.081 | 0.145 | 0.36 |
| *Eudaimonia* | FRED13 | 0.135 | 0.177 | 0.443 | 0.66 | 0.83 |
| | FRED15 | −0.511 | 0.126 | <0.001 | 0.349 | 0.16 |
| | FRED13 + 15 | −0.116 | 0.043 | 0.007 | 0.25 | 0.19 |
| $\delta_{hed-eud}$ | FRED13 | 0.401 | 0.331 | 0.225 | 1.132 | 0.73 |
| | FRED15 | 0.596 | 0.231 | 0.01 | 0.586 | 0.3 |
| | FRED13 + 15 | 0.189 | 0.079 | 0.017 | 0.346 | 0.26 |

$m \times m$ covariance matrix used to generate $Y$ using the rmvnorm function was a random sample of $S_Y$ each iteration. In each iteration of the simulation, the permutation and rotation null distributions were generated from 2,000 permuted samples. For the GEE and all OLS tests, 4,000 iterations were run for each value of $m$. For the GLS test, the number of iterations were 4,000, 2,000, and 1,000, for $m = 10$, 30, and 52.

Because of concern that the poor performance of the GLS method is due to misspecification of the error matrix, I re-ran the simulation (2,000 iterations) using an unstructured error matrix for $m = 10$. The simulation was not run for $m = 30$ and $m = 52$ because of problems of convergence. Regardless, the simulation at $m = 10$ was sufficient to show that the poor performance of the GLS method is not simply due to using a compound symmetry error matrix.

### Correlated estimation error

Regressors with a high positive correlation, as with *Hedonia* and *Eudaimonia*, have negatively correlated partial regression coefficients. I give a brief mathematical explanation of this in the discussion but also show this empirically using the Monte Carlo simulation.

All analyses were performed using R (*R Core Development Team, 2015*). All analysis scripts are included in Supplemental Information 1 and are available at the public GitHub repository https://github.com/middleprofessor/happiness.

## RESULTS

### GLS replication of previous analyses

The variance-standardized effects for hedonic and eudaimonic scores estimated from the GLS for the FRED13 and FRED15 datasets are given in Table 1. My estimates for FRED15 and the combined FRED13 + 15 are within 0.002 standard units of those reported in *Fredrickson et al. (2015)*. *Fredrickson et al. (2015)* do not report the GLS results for the 2013 data. My estimates of the coefficients for the FRED13 data show a pattern opposite to that for FRED15. That is, with the 2013 data, the effect of *Hedonia* is large and has a very small $p$-value (0.002) while the effect for *Eudaimonia* is small and not statistically significant ($p = 0.44$). My FRED13 coefficients are the same as those reported in the exploratory

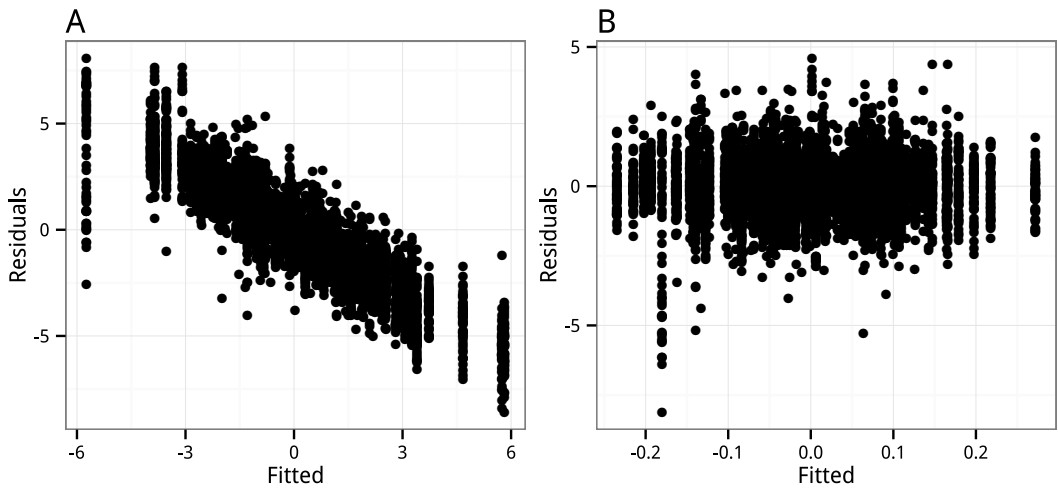

**Figure 1** Residual versus fitted values from the (A) fixed effects linear model with correlated error (GLS), and (B) Generalized Estimating Equations (GEE) model. The pattern in (A) indicates strongly biased estimates.

**Table 2** GLS estimates of the variance-standardized mean effect of *Eudaimonia* on CTRA gene expression for the FRED13, FRED15, and COLE15 data specifying a heterogenous compound symmetry error matrix. *Hedonia* was excluded from the FRED13 and FRED15 analyses and *hispanic* and *ln(hh.income)* were excluded from the COLE15 analysis so that all analyses had the same set of covariates.

| Data | $\bar{\beta}_{eudaimonia}$ | SE | $p$ |
|---|---|---|---|
| FRED13 | 0.558 | 0.107 | < 0.001 |
| FRED15 | −0.519 | 0.086 | < 0.001 |
| COLE15 | −0.007 | 0.076 | 0.931 |

re-analysis of the FRED13 and FRED15 datasets by *Brown et al. (2016)*, who also note the opposite pattern from the 2015 results. A diagnostic plot of residual versus fitted values from the GLS model for FRED15 suggests strongly biased estimates (Fig. 1A). Estimates for the effect of *Eudaimonia* for COLE15 are given in Table 2, with estimates from FRED13 and FRED15, with *Hedonia* excluded from the model, for comparison. The gene set effect of *Eudaimonia* for the COLE15 data is trivially negative and not statistically significant. By contrast, the effect of *Eudaimonia* is large and positive for FRED13 and large and negative for FRED15 (Table 2).

### New results

Standardized mean effects ($\bar{\beta}$) estimated by multivariate regression (OLS) are very small and positive for *Hedonia* and very small and negative for *Eudaimonia* for both 2013 and 2015 datasets and the combined dataset (Table 3). The bootstrap SE for each effect is too large, relative to the signal, to have any confidence in the direction of either of the effects for any dataset. This inference from the bootstrap SE is generally supported by the $p$-values of the OLS tests, with the exception that the $p$-values from the GlobalAncova and $F_{pun}$ tests for *Eudaimonia* for the combined (FRED13 + 15) dataset provide weak evidence of a negative effect of *Eudaimonia* on CTRA gene expression. The $p$-values from O'Brien's

**Table 3 OLS estimates of mean effects (Estimate) on CTRA gene expression.** The estimates are the mean variance-standardized partial regression coefficients from the multivariate regression over the $m$ responses (genes). $\delta_{\text{hed}-\text{eud}}$ is the difference in mean effect. The $p$-values are from O'Brien's OLS $t$-test, Anderson's $r_F^2$-test, GlobalAncova test, $F_{\text{pun}}$-test, and Roast.

| Type | Data | Estimate | $SE_{\text{obrien}}$ | $p_{O'\text{Brien}}$ | $p_{r2}$ | $p_{GA}$ | $p_{\text{pun}}$ | $p_{\text{Roast}}$ |
|---|---|---|---|---|---|---|---|---|
| *Hedonia* | FRED13 | 0.026 | 0.102 | 0.8 | 0.77 | 0.86 | 0.85 | 0.78 |
| | FRED15 | 0.062 | 0.052 | 0.23 | 0.22 | 0.28 | 0.28 | 0.23 |
| | FRED13 + 15 | 0.052 | 0.042 | 0.22 | 0.21 | 0.38 | 0.37 | 0.21 |
| *Eudaimonia* | FRED13 | −0.063 | 0.093 | 0.5 | 0.49 | 0.77 | 0.77 | 0.49 |
| | FRED15 | −0.067 | 0.052 | 0.2 | 0.19 | 0.11 | 0.11 | 0.2 |
| | FRED13 + 15 | −0.064 | 0.043 | 0.14 | 0.13 | 0.04 | 0.04 | 0.13 |
| $\delta_{\text{hed}-\text{eud}}$ | FRED13 | 0.089 | 0.19 | 0.64 | | | | 0.6 |
| | FRED15 | 0.129 | 0.104 | 0.22 | | | | 0.18 |
| | FRED13 + 15 | 0.116 | 0.085 | 0.17 | | | | 0.14 |

**Table 4 Generalized Estimating Equations estimates of the effects and difference in effects ($\delta_{\text{hed}-\text{eud}}$).** SE is a robust standard error.

| Type | Data | Estimate | SE | $p$ |
|---|---|---|---|---|
| *Hedonia* | FRED13 | 0.026 | 0.098 | 0.79 |
| | FRED15 | 0.062 | 0.04 | 0.12 |
| | FRED13 + 15 | 0.052 | 0.039 | 0.19 |
| *Eudaimonia* | FRED13 | −0.063 | 0.098 | 0.52 |
| | FRED15 | −0.067 | 0.046 | 0.14 |
| | FRED13 + 15 | −0.064 | 0.039 | 0.1 |
| $\delta_{\text{hed}-\text{eud}}$ | FRED13 | 0.089 | 0.189 | 0.64 |
| | FRED15 | 0.129 | 0.079 | 0.1 |
| | FRED13 + 15 | 0.116 | 0.073 | 0.11 |

OLS $t$-test, Anderson's $r_F^2$-test, and the Roast test are very similar to each other across all tests. Similarly, the $p$-values from the two permutation $F$-tests (GlobalAncova and $F_{\text{pun}}$) are very similar to each other across all tests. The GEE estimates are the same as the OLS estimates to the 3rd decimal place for all three datasets and the robust standard errors are large relative to the coefficients (Table 4). The GEE $p$-values are very similar to those from the OLS tests (especially the grouping of O'Brien's OLS $t$-test, Anderson's $r_F^2$-test, and the Roast test) for all three datasets and fail to reject any of the nulls. A diagnostic plot of residual versus fitted values from the GEE model for FRED15 does not suggest biased estimates (Fig. 1B).

The GLS bootstrap standard errors for FRED13, FRED15, and the combined data are given in Table 1. The bootstrap distributions of standardized effects for *Hedonia* and *Eudaimonia* for FRED15 are shown in Fig. 2. The bootstrap standard errors computed from these distributions are 2–4 $\times$ the parametric standard error. The bootstrap standard errors were computed with the predictor *Smoke* excluded from the model. This exclusion had only trivial effects on the standard error (Table S1). The GLS permutation $p$-values are high relative to the parametric $p$-values and fail to reject the nulls for any of the tests

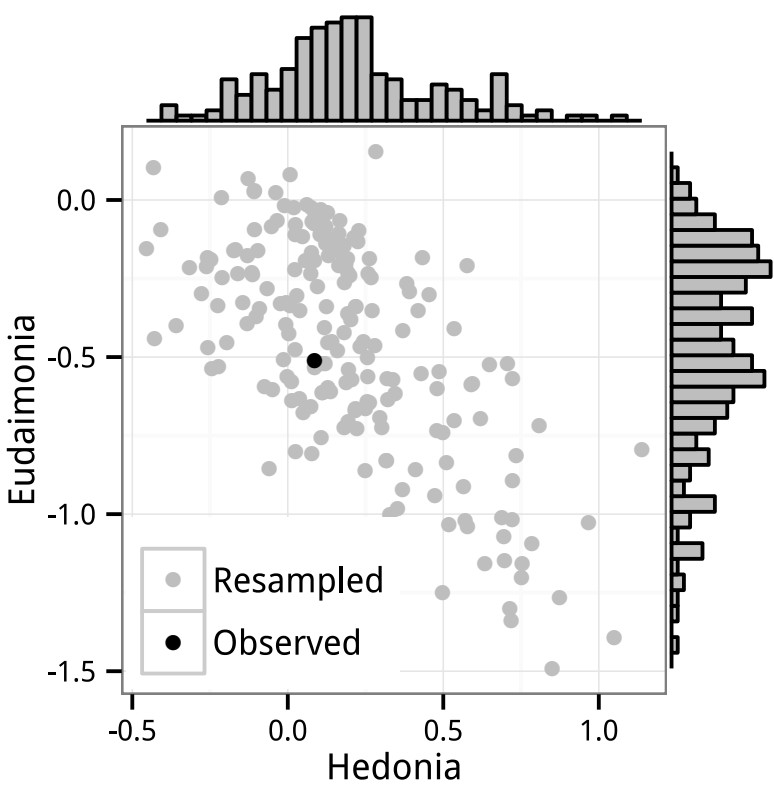

**Figure 2** **Distribution of GLS bootstrap resampled standard partial regression coefficients for *Hedonia* and *Eudaimonia*.** The data are the FRED15 dataset and the coefficients were estimated by the linear model with correlated error (GLS). Also shown is the observed value for FRED15(black).

at $\alpha = 0.05$ (Table 1). Compared to the GEE $p$-values, the GLS permutation $p$-values are much less similar to those from the OLS tests.

### Test size, power, sign error, and magnitude error

Type I error, power, sign, and magnitude error as a function of $m$ are shown in Fig. 3 and tabled in Table S2. The GLS test has inflated Type I error rates that increase with gene set size ($m$). At the gene set size of the FRED15 data ($m = 52$), Type I error is ∼28% for the GLS test (Fig. 3A). Type I error in the GLS using an unstructured error matrix (0.106) is almost identical to that when using a heterogenous compound symmetry error matrix (0.103) at $m = 10$ (the only value of $m$ tested because of problems of convergence with larger $m$). In contrast to the GLS results, Type I error for all alternative methods are relatively stable as $m$ increases. Notably, Type I error for all OLS tests are near the nominal level (0.05) while that for GEE is slightly elevated (∼0.08).

Power of all tests increases with gene set size ($m$) (Fig. 3B). GlobalAncova and $F_{\text{pun}}$-tests have ∼2× the power of O'Brien's, Anderson's $R_F^2$ and Roast tests when $m = 52$, without inflation of Type I error. GLS also has relatively high power when $m = 52$ but this is associated with the inflated Type I error. An adjusted Power was computed for GLS by finding the $\alpha$ that results in a Type I error rate of 0.05. At this adjusted $\alpha$, the power of the GLS is 0.16, 0.14, and 0.12 at $m = 10$, 30, and 52, which is substantially less than the power

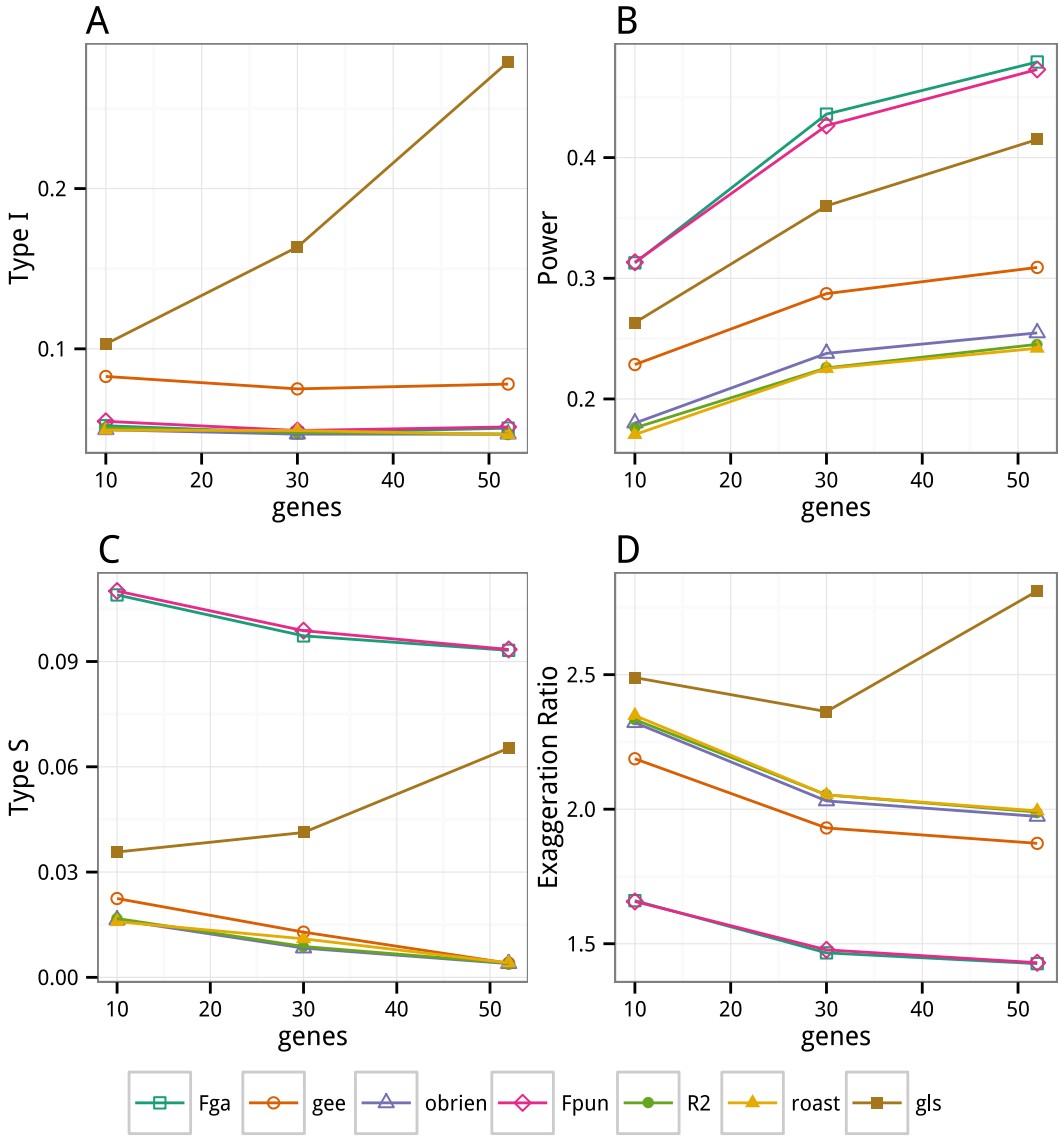

**Figure 3** **Errors for the different test methods based on Monte Carlo simulation of FRED15 dataset.** (A) Type I error when the true effect is zero. (B) Power when the true effect is 0.067 (the absolute value of the OLS estimated effect of *Eudaimonia* on mean expression for the FRED15 dataset). (C) Type S ("sign") error when the true effect is 0.067. Type S error is the fraction of statistically significant effects in which the estimate has the opposite sign of the true effect. (D) Type M ("magnification") error when the true effect is 0.067, illustrated by the Exaggeration Ratio (ER). ER is the ratio of the estimated to true effect when $p \leq 0.05$.

of the OLS tests. GEE has about $1.2\times$ the power of O'Brien's, Anderson's $R_F^2$, and Roast tests when $m = 52$, but at a small cost of Type I error.

The inflated Type I error and power of the GLS test is due not only to downward bias in the standard errors (see bootstrap results above) but also to an upwardly biased estimate of the mean effect. Using the Monte-Carlo Type II simulation results, the GLS estimate when $m = 52$ converges to a value that is biased upward by 30% (computed as $\frac{\bar{\beta}_{estimated} - \beta_{true}}{\beta_{true}}$, where the mean is taken over all 1,000 iterations).

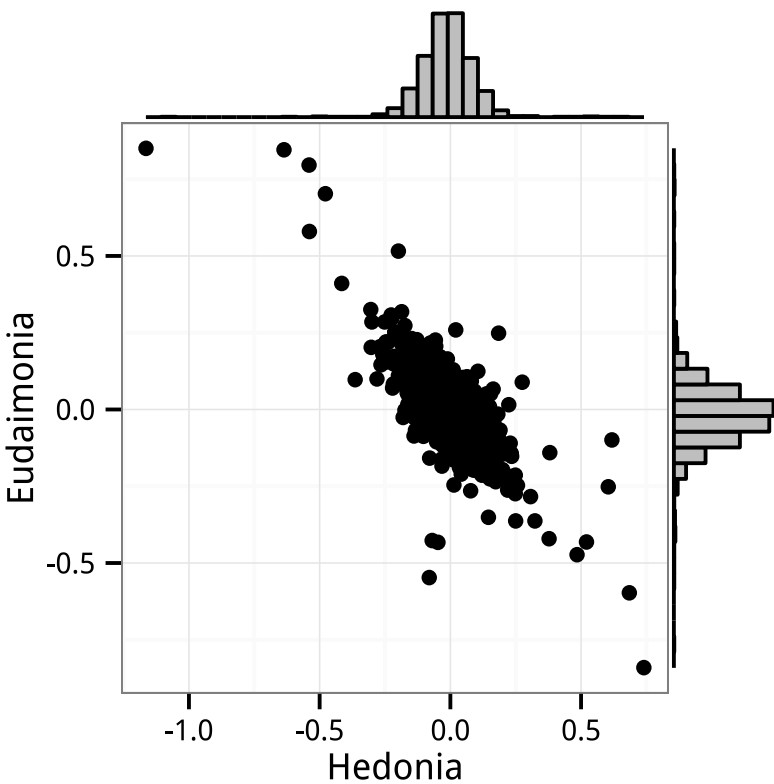

**Figure 4 Bivariate distribution of standard partial regression coefficients for *Hedonia* and *Eudaimonia* from the Monte Carlo experiments.** The Monte Carlo simulated the FRED15 data but with zero expected effect of any of the regressors on the gene expression levels. The coefficients were estimated by the linear model with correlated error (GLS).

With the exception of GLS, Type S error decreases with gene set size. At the gene set size of FRED15, Type S errors are trivially low (0.004) for GEE, O'Brien's, Anderson's $R_F^2$ and Roast tests, moderate (0.065) for GLS and high (0.093) for the permutation $F$ tests (Fig. 3C). The Exaggeration Ratio (ER) decreased with gene set size in all methods except GLS (Fig. 3D). As a consequence, at $m = 52$, statistically significant effect sizes estimated by GLS averaged 2.8× the true size. By contrast, when $m = 52$, statistically significant effect sizes estimated by GEE, O'Brien's, Anderson's $R_F^2$, and Roast were only 2× the true size, while the ERs for GlobalAncova and $F_{\text{pun}}$-tests are ∼1.4.

## Correlated coefficients

The expected, large negative correlation between the partial regression coefficients for *Hedonia* and *Eudaimonia* are shown using the GLS bootstrap distribution (Fig. 2) and using the GLS Monte Carlo simulation results (Fig. 4). Despite modeling the empirical correlations among the regressors and among the response variables, the distribution of standardized coefficients from the GLS Monte Carlo simulation have a much smaller range than that from the GLS permutation (e.g., 95% confidence interval for $\beta_{eudaimonia}$ from the Monte Carlo simulation is −0.20 to 0.23 while that from the GLS permutation is −0.62 to 0.57), which suggests there is something about the structure of the actual data that is inflating the coefficient estimates (*Littell et al., 2006*).

## DISCUSSION

A causal association between well-being components and CTRA expression levels would be an important discovery. Certainly, some association between well-being scores and CTRA expression levels must exist because of common shared paths within the complex network of causal paths of the underlying physiology. Nevertheless, observational studies like that of *Fredrickson et al. (2013)* and *Fredrickson et al. (2015)* are poor designs for discovering knowledge (*Walker, 2014*). The re-analysis of the CTRA gene expression data in subjects scored for hedonic and eudaimonic well-being highlights several important results: (1) any effect of hedonic and eudaimonic well-being on CTRA expression is very small and the noise is too large relative to the signal to reliably estimate the sign and magnitude of these mean effects; (2) the apparent replication of opposing effects is most parsimoniously explained by correlated noise due to the high correlation between *Hedonia* and *Eudaimonia*; (3) the GLS with correlated error test has high error rates and inflated effect estimates for simulated data modeled on the focal dataset; and (4) all of the OLS tests have appropriate error rates and the permutation $F$-tests have high power.

### The association between well-being and CTRA expression

Standardized mean effects of *Hedonia* and *Eudaimonia* are very small (Table 3) but the standardization effectively precludes easy comparison to published effect sizes on expression levels. The multivariate (OLS) regression was re-run on the unstandardized expression levels of FRED13 and FRED15 and the mean coefficients were back-transformed to a fold change per four standard deviation change in the predictor (effectively comparing someone at the high and low ends of the well-being axis), using $FC = 2^{4*\bar{\beta}}$. For *Eudaimonia*, I used the reciprocal of this fold change to make the value greater than one and multiplied it by $-1$ to indicate a decreasing effect. The FC values were 1.036 and 1.072 for *Hedonia* and $-1.078$ and $-1.06$ for *Eudaimonia* (note that *Fredrickson et al. (2015)* reported this fold change as a percent). The biological significance of such a small mean effect awaits experimental evidence.

Several features of the GLS results suggest unstable and inflated coefficient estimates resulting from the GLS model. First, the highly variable pattern of effects among the three datasets (FRED13, FRED15, COLE15) when estimated using the same error structure suggests that either a large lack of generalizability among samples or the coefficients are more unstable than suggested by their (non-robust) standard error. Second, the GLS coefficients are very different from and generally much larger than the OLS coefficients (Tables 1 and 3). Third, at least one of the GLS coefficients in each of the datasets is very large relative to what we would expect from a gene set association given observational data and the stated hypotheses. Fourth, in a supplementary table, *Fredrickson et al. (2015)* report strikingly different results (small, negative coefficients for both *Hedonia* and *Eudaimonia*) for the FRED15 dataset using an unstructured error matrix for the GLS computation (*Hedonia* : $\beta = -0.014, p = 0.17$; *Eudaimonia* : $-0.0026, p = 0.81$. Compare these to Table 1). *Fredrickson et al. (2015)* failed to identify or address any of these concerns, including why the 2013 dataset was not analyzed using the updated (GLS) analysis, or how to interpret the differing results using a compound symmetry error matrix, which was

the focus of *Fredrickson et al. (2015)*, or an unstructured error matrix, which was the focus of *Cole et al. (2015)*. The results reported here support the conclusion of biased and inflated coefficient estimates from the GLS. These results include the large coefficients that commonly occurred in the GLS with permuted data despite the expected effects of zero and the diagnostic plot of the residual vs. fitted values that indicates biased estimates (Fig. 1).

## The replication in the pattern of effects between datasets

The apparent replication of opposing signs for hedonic and eudaimonic effects on CTRA expression (*Fredrickson et al., 2015*; *Fredrickson et al., 2016*) can be inferred only from the OLS estimates; the GLS estimates are strikingly inconsistent with a replicated pattern of expression. This failure of the GLS estimates to replicate was not noted by *Fredrickson et al. (2015)* because they used the OLS estimates to illustrate the replication but GLS to infer effects. Regardless, any replication in the sign of the mean effect should not be surprising given only two replicates of two coefficients.

The pattern of opposing signs for hedonic and eudaimonic effects on CTRA expression is consistent with very small effects in combination with the high empirical correlation between hedonic and eudaimonic scores (0.80 in FRED13 and 0.74 in FRED15). Partial regression coefficients of regressors that are positively correlated are themselves negatively correlated because their estimation shares common components that are of opposite sign. This is easily shown using the data from FRED15 where, disregarding all predictors but hedonic and eudaimonic scores, the partial regression coefficient of any gene expression level on *Hedonia* ($X_1$) and *Eudaimonoia* ($X_2$) are

$$\beta_1 = 0.018 x_1^\top y - 0.013 x_2^\top y \\ \beta_2 = -0.013 x_1^\top y + 0.018 x_2^\top y \tag{6}$$

where the 0.018 and $-0.013$ are the diagonal and off-diagonal elements of the inverse of the $X^\top X$ matrix of FRED15 (again disregarding all other predictors to simplify the explanation). Because of the high correlation between hedonic and eudaimonic scores, both $\beta_1$ and $\beta_2$ include a large contribution from the covariance of the other $X$ with $Y$ but the sign of this contribution is negative. Consequently, if the true effects are trivially small, then the pair of $\beta$ coefficients will tend to have opposite signs because of the negative correlation of estimates centered near zero. Random noise creates negatively correlated coefficients that tend to be opposite in sign. Linear mixed models do not adjust for this correlation. The negative correlation between coefficients is easily seen in the distribution of bootstrap GLS estimates of $\beta_{hedonia}$ and $\beta_{eudaimonia}$ (Fig. 2). The tendency for the coefficients to have opposite signs if the expected effects are zero is seen in the Monte Carlo simulation of the FRED15 data (Fig. 4). While I have shown the negative correlation using GLS estimates, this correlation would also appear in OLS estimates. The most parsimonious explanation of the apparent replication of opposing effects of hedonic and eudaimonic scores on CTRA gene expression is correlated noise arising from the geometry of multiple regression.

## Comparison of method performance

The Monte-Carlo simulations of the GLS with correlated error for repeated measures or multiple outcome data are consistent with other studies demonstrating inflated Type I

error due to downward biased standard errors (*Guerin & Stroup, 2000*; *Littell et al., 2006*; *Jacqmin-Gadda et al., 2007*; *Gurka, Edwards & Muller, 2011*). Importantly, the inflated Type I error of the GLS is not simply due to failure to specify an unstructured error matrix as the Type I error when $m = 10$ is nearly the same regardless of which error matrix is specified. In contrast to the GLS tests, all of the OLS tests maintain error rates close to the expected value (0.05). The permutation $F$-tests ($F_{pun}$ and GlobalAncova) have much higher power than the O'Brien's OLS, Anderson's $r_F^2$, and Roast tests and, unlike the moderately high power for GLS, this power does not trade-off with Type I error.

In designs with low power because of small effect sizes, Type S and M errors are more likely to emerge (*Gelman & Carlin, 2014*). That is, with low power, only unusually large estimates are large enough to reach statistical significance. And with a true effect size near zero, an estimate with unusually large error from the true value has a high probability of being the wrong sign. In the simulation here, the true effect is small but the tests with the highest power are associated with the highest rate of Type S error. Sign error is a cost of a higher powered test. This Type S error affects the permutation $F$-tests, which have $10\times$ the Type S error as the other OLS tests. Indeed, Type S error, even with a very small effect, is trivial in the O'Brien's, Anderson's, and Roast tests. The exaggeration ratio (ER), a measure of Type M error (*Gelman & Carlin, 2014*), is a good indicator of the expected inflation of an estimate when the design or test has low power. The expected inflation is nearly $3\times$ the true value for the GLS estimate under the conditions of the FRED15 dataset. By contrast, the expected inflation is less than $1.4\times$ for the $F_{pun}$ and GlobalAncova tests. The high powered tests result in the (perhaps paradoxical) negative relationship between Type M and Type S error among the tests.

## CONCLUSIONS

The OLS estimates combined with the permutation $F$-tests provide some evidence of a very small negative association between *Eudaimonia* and mean CTRA expression, although the Monte Carlo results of these $F$ tests raise some concern about the sign of this effect. As I have stated above, however, there must be some association between well-being components and CTRA expression, so an observational design with a statistically significant $p$-value should not cause much excitement. What we want to know are the causal pathways that explain this association—does decreased CTRA cause eudaimonic well-being, or does eudaimonic well-being cause decreased CTRA, or is the correlation jointly determined by an unknown causal pathway? Also, we want to know if the effect magnitude has meaningful physiological consequences.

The linear model with correlated errors (GLS) has few merits for the estimation of mean fixed effects across multiple responses. The estimation is time consuming and estimation with an unstructured error matrix is plagued with difficulties in convergence. Simulations of the model (here and elsewhere) with repeated measures or multiple outcomes show a high frequency of inflated coefficient estimates and downward biased standard errors. As expected, the Generalized Estimating Equations with robust standard errors perform much better than the GLS, but even this estimator has inflated Type I error. While all the OLS

methods maintain Type I error at the nominal rate, the tests using the $F$-ratio ($F_{pun}$ and GlobalAncova) have a relatively high power and small exaggeration ratio. A concern of the GlobalAncova test for observational data is the violation of the exchangeability assumption. How GlobalAncova performs relative to the $F_{pun}$-test with simulated data with moderate to large correlations between multiple predictors and nuisance covariates remains to be investigated.

## ACKNOWLEDGEMENTS

I am grateful to three reviewers and especially the associate editor for greatly improving this manuscript.

### Funding
The author received no funding for this work.

### Competing Interests
The author declares there are no competing interests.

### Author Contributions
- Jeffrey A. Walker conceived and designed the experiments, performed the experiments, analyzed the data, contributed reagents/materials/analysis tools, wrote the paper, prepared figures and/or tables, reviewed drafts of the paper.

### Data Availability
GitHub: https://github.com/middleprofessor/happiness.

### Supplemental Information
Supplemental information for this article can be found online at http://dx.doi.org/10.7717/peerj.2575#supplemental-information.

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
