# Peer review of "Monte Carlo simulation of OLS and linear mixed model inference of phenotypic effects on gene expression"

_PeerJ, doi:10.7717/peerj.2575_

## Round 0.1 · original submission · Minor Revisions

I am sorry for the long delay in giving you’re a decision on this manuscript. I fear that the combination of psychological content and highly technical statistics made it difficult to find reviewers. As you will see, the first two reviewers could not be more different in their evaluation. Reviewer 2 argued that your analysis and conclusions are flawed, whereas reviewer 1 praises the paper as clear and convincing. Reviewer 3 has some minor criticisms of the paper, but recommends minor revisions are sufficient deal with these.

To deal first with what reviewer 2 regards as the 'most significant issue', I disagree with their evaluation that your ms does not meet PeerJ criteria. There is precedent for PeerJ publishing analyses of simulated datasets (as in your Monte Carlo simulation of GLS) and also for new analysis of existing datasets, which are also presented in your ms. These contributions are novel enough to come under the scope of the journal. I do not regard your lack of track record in publishing psychology or genomics as relevant, given that the points you raise have to do with statistical methods. I also was puzzled by the statement that you had provided insufficient information about exact procedures performed. I had no difficulty in downloading these from Github, and found this information helpful for understanding what you had done. I am not sure whether there are problems of longevity with Github, but if that is the case, then I would recommend that you upload the scripts to another depository which could provide you with a long-term link and DOI. (I think there are several options: I am familiar with Figshare, Open Science Framework and Dryad).

I am not expert enough in biostatistics to adjudicate on all the points raised by reviewer 2, but I note that reviewers 1 and 3 have considerable expertise in this area and are satisfied with the basic analysis.

Nevertheless, reviewers 2 and 3 both raise concerns about whether you have entirely accurately characterised analyses conducted by Fredrickson et al 2015, and, clearly, this needs to be dealt with in a revision. Reviewers 2 and 3 also argue it is not correct to say that the effects of Fredrickson et al are due to correlated noise, when what you have shown is that simulated data can give similar effects. It is a fine distinction but would merit some rewording.

My impression is that the points you have raised about the analysis have broader application than to the specific papers by Fredrickson and colleagues that are your current focus. In particular the type I error inflation when GLS is applied to simulated correlated data is notable – and worrying in a time when many scientists are starting to use complex mixed effect models without necessarily appreciating the dependence of results on underlying assumptions. I suspect that there are many situations when this method is applied to data where correlated errors could be an issue, but many will not attract attention because they are not reporting such dramatic and high profile findings as Fredrickson et al.

I therefore suggest that this submission might have more impact if it were recast as a paper that is a more general critique of the problems that can arise with GLS when sample sizes are small and correlated variables are used. The analyses you have done could be used to illustrate that more general point, and to highlight the possible advantages of alternative methods. In particular, I think it worth recommending to anyone who plans to use these methods to apply them to simulated data with a similar structure to clarify whether the assumptions are valid. All too often it is hard even for statistically sophisticated researchers to judge this a priori. If you felt that was a good way to proceed, it would make the paper more generally valuable by recommending ways to avoid problems rather than just providing a critique of one study.

I do not insist that you take that approach, and think the paper could be accepted with the existing focus, provided you address the specific reviewer comments, either by modifying the ms or by explaining why you have not done so. However, I think that revising the paper to be more general would not require a great deal of further modification but it would extend the interest to a broader readership. A more generic title would then be appropriate.

Thank you for considering PeerJ as an outlet for your work. I look forward to seeing your revision.

·

Basic reporting

Very well written paper. Well structured. Although maybe a bit long: the arguments that the FRED2015 analysis is faulty would still be highly convincing with only half the evidence now presented by the authors.

I could not find the sample size of the FRED2013 paper. It would be helpful to have that mentioned.

Experimental design

The analyses and results are clear and convincing. Perhaps it could be mentioned that the ROAST analysis is small sample exact, whereas the permutation approach of Freedman and Lane is only asymptotically valid.

Validity of the findings

No comments Crystal clear.

Comments for the author

I am happy that the author has taken up such a thorough reanalysis of a published paper. Unfortunately there are many more papers out there that are in need of the same treatment. Still even a few papers like the submitted work will help raise methodological standards in the field.

Reviewer 2 ·

Basic reporting

The author of the current article concedes that he lacks expertise in psychology, and a Google Scholar search suggests that he has not published in the area of genomics either. Nevertheless, he proceeds to reanalyze the data from Fredrickson et al. (2013) and one of the two studies presented in Fredrickson et al. (2015). The article reports the results of four non-parametric methods. A weakness is that the author proceeds without justifying why non-parametric approaches are superior to the parametric approaches used in Fredrickson et al. (2013, 2015).

Experimental design

Because the manuscript is highly technical, I solicited input from a statistical colleague who provided these remarks:

"This article re-analyzes data from 2 previously published datasets using different analytic methods and finds different results (no statistically significant association between a specified gene expression profile and measures of hedonic well-being, eudaimonic well-being, or the difference between the two). The results are provocative and continue a persistent debate about the validity of results reported by Fredrickson et al in their cited 2013 and 2015 papers. However, the current manuscript is deficient on several PeerJ publication criteria and may not even fall within the publication scope of the journal.

The most significant issue involves PeerJ editorial criterion 2, as this paper does not 'describe original primary research.' All data analyzed here appear to have been published previously elsewhere, and the present manuscript reports only additional secondary analyses of those existing data. The present analyses appear to represent a step backward in terms of analytic rigor, so they would not seem to constitute any distinctive methodological contribution either. According to my reading of the PeerJ Aims & Scope, this contribution might be more appropriately treated as a technical commentary in PeerJ Preprints.

Given the above, the rest of my comments are probably moot. But even if this contribution is judged within PeerJ scope, it suffers from significant deficiencies on other editorial criteria involving technical rigor and accuracy of conclusions.

The article fails PeerJ editorial criteria regarding rigor / high technical standard / statistical soundness. Although the author provides extensive additional analyses with alternative methods, he never provides any clear reason to doubt the validity of the Fredrickson et al. (2015) mixed model results (and my read below is that they are more sound and appropriate than the new results reported here). The author concludes that the multiple non-significant findings from his analyses indicate the absence of a true association between well-being and the gene expression profile. However, the non-significant findings may be equally well explained by the substitution of less efficient non-parametric statistical analyses for the Fredrickson et al. (2015) mixed model analysis. This problem is compounded by potential biases arising from this author’s apparent specification of an exchangeable covariance structure for his analyses. The exchangeability assumption is contradicted by marked heterogeneity in both variance and covariance across the multiple genes analyzed (which both Fredrickson et al. and this author note, but only the former address by using more appropriate error structure specifications). Given that this paper uses less powerful and less appropriate statistical models, is no surprise that it yields different and weaker results. The non-significant results observed here may reflect more about the data analyst’s choice of less efficient and accurate analytic methods rather than anything about the substantive data. Loss of power and precision would be observed in any analysis that substituted nonparametric techniques for parametric models when the latter are appropriate.

This article also fails PeerJ editorial criterion 2 in providing insufficient information about the exact procedures performed. At a minimum, this article needs to include a transcript of all statistical code (e.g., in appendix or supporting information file) so readers can better understand exactly what has been done, how it differs from other analyses, and why. (github is not sufficient because those postings often disappear)

Concerns about technical rigor/statistical soundness are also raised by multiple presentational errors or misrepresentations, detailed below. For example, the current author (1) characterizes the Fredrickson et al. (2015) analysis as a 'GLS' 'marginal model' when in fact Fredrickson et al. (2015) report using a standard conditional mixed linear model estimated by maximum likelihood; (2) misrepresents the number of data points analyzed as n = 122 or 198 rather than the 6,344 and 10,372 observations actually analyzed (which provide ample data for estimating the mixed effect models, contra the author’s claim); (3) asserts that marginal effects are 'typically' estimated by GEE, even though mixed model analyses are also widely used for this purpose, perfectly valid for it, and often more powerful (particularly in the present case involving mis-specified error structure for the GEE); (4) asserts that Fredrickson et al. 2013 and 2015 failed to test the 'delta' hypothesis of the difference (eudaimonic – hedonic) when I was able to find those results easily in both cases; (5) states that he’s reproduced the Fredrickson et al. analysis results when differences are apparent (i.e., in his Table 2); (6) asserts that the difference in his Table 2 results vs. Fredrickson et al. (2015) is due to variations in the posted data (I verified that it was not, and speculate that the difference instead arose from variations in model specification or estimation, such as use of GLS instead of maximum likelihood, compound symmetry rather than more appropriate error structures, etc.); (7) concludes the difference between bootstrap SE’s and parametric model SE’s stems from models’ 'sensitivity' to sampling variability (rather than to the nonparametric nature of the estimation algorithm and the well-known problem of bootstrap dilation (e.g., Bradley Efron (2010) Correlated z-Values and the Accuracy of Large-Scale Statistical Estimates, Journal of the American Statistical Association, 105:491); (8) in analyzing Type I error in the GLS method, produces Monte Carlo simulations that don’t correspond to the characteristics of the original analysis they are subsequently used to critique; (9) states that 'Random noise creates negatively correlated error' (which may be true for his Monte Carlo data, but is a nonsensical statement for empirical data analyzed by a different model; moreover all sources of parameter correlation seem to have been appropriately accounted for in the SE estimates for the mixed model). More misrepresentations could be cited. These inaccuracies left it difficult for me to trust any of the remaining assertions without verifying them myself. In the several cases where I did so, I noted a pattern of selective reporting and interpretation. For example, when I attempted to reproduce the author’s preferred GEE analyses, I found residuals showing much greater deviations from normality and poorer fit than those of the Fredrickson et al. (2015) mixed effect models. In general, these analyses appear to move us farther away from analytic validity."

Validity of the findings

Again, from my statistical colleague:

"The contribution fails PeerJ editorial criterion 3 by asserting substantive conclusions well beyond what could possibly be supported by the analytic results. Perhaps the most significant interpretive error is the author’s overall conclusion that the pattern of gene expression is 'simply correlated noise.' This conclusion commits the elementary statistical error of accepting the null hypothesis (see Goodman S. A dirty dozen: twelve p-value misconceptions. Semin Hematol. 2008 Jul;45(3):135-40. PubMed PMID: 18582619). If the author didn’t generate the data (e.g., by simulation) then it is impossible to say how the associations actually arose. At minimum the conclusions need to comply with the recent ASA consensus statement on appropriate interpretation of p-values (http://amstat.tandfonline.com/doi/pdf/10.1080/00031305.2016.1154108).
The most the author could validly claim is that the magnitude of associations estimated in his specific nonparametric analysis of the Fredrickson et al. data are not inconsistent with correlated noise as specified in that model. To meet the criterion of reporting all relevant results, the Discussion needs to note that the data are clearly inconsistent with correlated noise under the mixed model specification applied by Fredrickson et al. (2015). An accurate interpretation of the present results would also note that the non-significant results observed here may result from the comparative inefficiency of nonparametric techniques used here, rather than from the absence of a true relationship in the population data. The ultimate question is whether the GEE or mixed model should be preferred on epistemological or goodness-of-fit grounds. Given the inconsistency of current nonparametric results with those of Fredrickson et al. (2013, 2015), it is also inaccurate to claim that results are 'unambiguous.' The primary conclusion is inaccurate on multiple grounds and greatly over-reaches what could ever be known from any secondary analysis of empirical data."

Reviewer 3 ·

Basic reporting

The article is clear and understandable.

Experimental design

This is an interesting study that uses a variety of techniques to investigate the relationships between CTRA gene expression and measures of happiness. Overall, the statistical work seems appropriate to me, and extends and applies mostly well-known statistical techniques – namely a number of permutation and bootstrap approaches. The first bootstraps data rows, the second is a residual permutation procedure to generate a null (but retain the ability to use covariates), the third is similar with a different test statistic, and the fourth involves a rotation test, using a procedure that I am less familiar with. The fifth and sixth are perhaps the most critical – they use the same procedure as Frederickson et al. (2015) but with bootstrapped, or permuted, data, and so test their procedure fairly directly. All of these approaches share a similar driving motivation, which is to test the robustness of the procedures used in two previous published papers, one of which attracted some previous critical responses, and the resulting conclusion.
Alongside these analyses of the real data, there is also a more limited analysis of simulated gene-expression like data, to illustrate the fact that the GLS procedure can inflate type I error, when untrue modelling assumptions are made (particularly regarding the covariance matrix S_Y).

Validity of the findings

Overall I have no serious arguments with the methods used or the conclusions reached.

The bootstrap and permutation approaches both imply that GLS estimates of errors are substantively underestimated in these data, and correcting for this completely alters the conclusions in terms of result significance. Because individuals are bootstrapped wholesale, the (unknown) covariance structure of the data is conserved. The principle aim here is to estimate uncertainties of parameter estimates, using an approach robust to likely causes of model mis-specification.

Comments for the author

I do have some specific points:

1) The null of no effect in either happiness measure, beta_hedonia=beta_ Eudaimonia=0 is a natural one for a non-specialist to consider as a start point – can the author explain why this is not of interest, to aid a general reader – or simply include this test and comment briefly? I believe the previous studies also investigated this question, via an F-test. This is natural as the authors say their aim is to study “what is the evidence for effects of hedonic and eudaimonic happiness scores on CTRA gene expression”. Then separately setting beta_hedonia=0 and beta_ Eudaimonia=0 (or indeed the same) become natural next questions, to evaluate whether one can separate the two coefficients.
2) In the abstract: “is simply correlated noise”. It is not possible to be statistically certain of this, given only a failure to reject the null hypothesis. It would be better to write “is consistent with correlated noise” or similar.
3) “This small sample per parameter ratio is likely to result in overfit models, which, in turn, will result in unstable and inflated coefficients 78 (Harrell, 2015).” This seems a somewhat unjustified statement to me, because in fact there are 122/198 observations per gene. For each gene, only a small number of parameters must be estimated. The deeper problem raised here is the potentially highly complex correlation structure in the underlying data. This is a serious potential confounder, and would (should) result in a very large number of parameters to fit relative to the data size, resulting in the previous studies using what are likely over-simplified, rather than overly complex, models.
4) Following on from (3), I note that in the recent Fredrickson et al. study (2015), their Table S3 tests the same hypotheses under a general covariance structure, with a mixed effect linear model. This does not seem to be discussed in the Walker manuscript, but should be critically discussed – because the manuscript currently implies that the compound symmetry covariance structure was the only one used. In the Fredrickson study their results for several measures change substantively when using the more general model, although they suggest convergence issues might be at fault.
5) “Finally, and most importantly, the GLS model gains power by assuming that the estimated regression coefficient is a common effect for all genes”. The current wording reads as if this point potentially invalidates the analysis of Frederickson et al. – because the common effect assumption is likely not to be true. However, what is important for the argument of the present study is whether this invalidates the test (rather than altering its power – indeed increasing the power is a favourable outcome if true). Because the null hypothesis is that all effect sizes are (equal and) zero, an alternative of equal non-zero effect sizes results in a valid test in principle - so I think this criticism should be withdrawn. Indeed, if effect sizes are heterogeneous this would reduce the power, but not increase the type I error rate, of the GLS test. Again, the point here is really that the previous authors used an overly simplified model, therefore not robust to departures from modelling assumptions.
6) In the recent Fredrickson et al. study, I believe they do actually test for a difference in coefficient of Hedonia or Eudaimonia (at least in a sense). E.g. from their paper: “In direct comparison of standardized association coefficients, the magnitude of CTRA association with eudaimonic scores significantly exceeded that for hedonic scores (t(104) = -2.59, p =. 0109).” This seems to be a t-test for a difference in coefficients, but is described here as a test of p-values – language which I think is misleading and should be explained more clearly in terms of its difference with the procedure here - i.e. the coefficients are standardised before comparison in the Fredrickson et al. case, but not in the Walker case, as I read it.
7) On p3 “( ¯βhedonia and ¯ Bhedonia) is a typo (repeated).
8) It would be helpful to include evidence of model mis-specification that underlies the identified problems with the GLS approach here - for example the behaviour of the observed residuals (discussed briefly) could be shown, or tested relative to their expectations.

---

## Round 0.2 · Minor Revisions

Dear Dr Walker

Thank you for your revised manuscript, and for taking the trouble to make further revisions. I find your responses to reviewers convincing and think the paper makes an important methodological contribution to the field.

However, I remain concerned about accessibility of the material to the less specialised reader. As I mentioned in my previous letter, I think the analytic approaches you cover, and the points you make about their strengths and limitations, have greater applicability than to the specific datasets of Frederickson and colleagues, but it is unlikely that you will get a broad readership unless a bit more is done to make it easier for people to approach this.

I really wanted to understand the statistical methods and the differences between them, so I turned to your R script. Alas, it was extremely hard to follow – largely because it was a faithful reproduction of what you had done to create the analyses of the Frederickson et al data. I’m sure we do need such a script for the record, but for someone who just wants to work through the different types of analysis, this feature, coupled with the fact that the sequence of analyses in the script does not follow that in the paper, makes it extremely daunting.

In the interests of trying to understand it, I created a pruned script which also included code for generating a simulated dataset with a smaller N variables. This is going well beyond usual editorial behaviour, but I did really want to understand this – in part so I could make a confident editorial judgement, and in part because I can anticipate using some of these approaches in future. I am not an expert in either programming or statistics, but I suspect my level of knowledge is similar to many people who might be interested in your paper. These are people who really appreciate being able to work through a script, rather than just being confronted with the formulae.
I will attach the script I ended up with (NB the PeerJ system only allows me to attach pdfs, so I will try also to deposit this on Github). I gave up at the point of generating the tables, because the naming of outputs from the statistical functions did not seem to match the names required for the table creation. I have some understanding of matrix algebra in R, but I could not work out how to compute the O’Brien’s OLS t-test from the variables created in the script. In addition, two of the statistical functions, limma and GlobalAncova were not available for my version of R (3.3), so I have not included the analyses depending on them. I don’t want to create a huge load more work for you – you have already put in a heroic amount of effort to do a very thorough job. But if you could include with the paper a much shorter and more generic script that just focuses on the principal analyses, without requiring modifications for the specifics of the Frederickson datasets, I think that would make the paper much more valuable to readers, as it would allow them to apply the methods to their own datasets. The script should be documented so that the reader can relate the analyses to the specific statistics reported in the paper.

There is a question of where to deposit the scripts. Given that reviewer 2 had raised queries about permanence of Github, I asked the PeerJ staff about this, and they responded: “the authors could certainly point to GitHub as the working repository for the most up to date version, but the ideal would be that the (current) script is also deposited somewhere 'archival'. From that point of view, they can supply it as a Supplemental File (i.e. attached to the article) or, if the script is really too large, then they could post it on Figshare or Data Dryad perhaps.’

Apart from this issue, my other comments are pretty trivial and should take little time to address.

Title: I wonder about just calling this “Monte Carlo simulation of OLS and linear mixed model inference of phenotypic effects on gene expression”

I see no problem with keeping the Abstract as it is, and including relevant keywords so those interested in the Frederickson paper will find this paper. My concern is rather that you may reduce the number of people who look at the paper because, for those outside psychology, eudaimonic and hedonic well-being are not familiar concepts and they do sound rather niche. And, as I noted previously, I think the implications of your work extend beyond Fredrickson et al, as increasingly non-geneticists are able to get measures of gene expression to relate to phenotypes and it is easy for them to apply inappropriate statistical models. Your paper emphasises the range of approaches that have been used and give appropriate and converging results, and I think potentially have broad application.
Minor comment:I found it rather distracting that you had italicised Hedonia and Eudaimonia – is that normal for phenotypes in gene expression studies? If so, then do retain this, but it made them look more like species or genes to me.

line 70. It would help if you would briefly define eudaimonic and hedonic well-being, as these are highly specialised terms. Could also mention these are measured by self-report questionnaire and are moderately intercorrelated.

lin 74, rather than ‘replicated’ in quotes, which is a bit ambiguous, you could say ‘were in the same direction as’

line 83, define BMI
line 197 ‘ “was whether”
line 274, typo “increasing” twice
line 328, ‘increase’
line 381, 415, 442: avoid contractions as too informal: eg ‘we would’ rather than ‘we’d’
line 457; typo “violation”
line 461; “reviewers”
Reference formatting
I think the correct formatting for PLOS One is without capitalisation of the One.
typo in Gelman and Carlin title
inadequate reference for Guerin and Stroup – needs to be in public domain
Ritchie et al, need to capitalise initial word of title

---

## Round 0.3 · accepted · Accept

Many thanks for all your patience in the processing of this paper. I think the end result will make an important contribution to the field, and the accessible scripts will be welcomed by other readers.